# How Different Dietary Methionine Sources Could Modulate the Hepatic Metabolism in Rainbow Trout?

Chiara Ceccotti [1] , Ilaria Biasato [2] , Laura Gasco [2] , Christian Caimi [2] , Sara Bellezza Oddon [2], Simona Rimoldi [1] , Fabio Brambilla [3] and Genciana Terova [1],*

[1] Department of Biotechnology and Life Sciences, University of Insubria, 21100 Varese, Italy; chiara.ceccotti@uninsubria.it (C.C.); simona.rimoldi@uninsubria.it (S.R.)

[2] Department of Agricultural, Forest and Food Sciences, University of Turin, 10124 Grugliasco, Italy; ilaria.biasato@unito.it (I.B.); laura.gasco@unito.it (L.G.); christian.caimi@unito.it (C.C.); sara.bellezzaoddon@unito.it (S.B.O.)

[3] VRM S.r.l. Naturalleva, 37044 Cologna Veneta, Italy; fabio_brambilla@naturalleva.it

* Correspondence: genciana.terova@uninsubria.it

**Abstract:** In aquafeeds in which plant proteins are used to replace fishmeal, exogenous methionine (Met) sources are demanded to balance the amino acid composition of diets and meet the metabolic fish requirements. Nonetheless, since different synthetic Met sources are commercially available, it is important to determine their bioavailability and efficacy. To address this issue, we conducted a two-month feeding trial with rainbow trout (*Oncorhynchus mykiss*), which were fed diets supplemented with five different forms of Met: Met-Met, L-Met, HMTBa, DL-Met, and Co DL-Met. No differences in growth performance were found in trout fed with different Met forms, but changes in the whole-body composition were found. In particular, Met-Met and L-Met promoted a significant body lipid reduction, whereas the protein retention was significantly increased in fish fed with HMTBa and Co DL-Met. The latter affected the hepatic Met metabolism promoting the trans-sulfuration pathway through the upregulation of *CBS* gene expression. Similarly, the L-Met enhanced the remethylation pathway through an increase in *BHMT* gene expression to maintain the cellular demand for Met. Altogether, our findings suggest an optimal dietary intake of all tested Met sources with similar promoting effects on fish growth and hepatic Met metabolism. Nevertheless, the mechanisms underlying these effects warrant further investigation.

**Keywords:** rainbow trout; methionine; liver; methionine hydroxy analogue; CBS; BHMT; SAHH; SAM; SAH

## 1. Introduction

Aquafeeds supply all the nutrients (protein, fats, carbohydrates, minerals, and vitamins) necessary for the optimal growth and health of cultured fish. Among nutrients, protein is the most expensive part of fish diets and supplies amino acids (AA) for energy and growth, and substrates for key metabolic pathways.

In aquafeed formulations, fishmeal (FM) has traditionally been chosen as the primary natural protein source for farmed fish due to its high nutritional value and digestibility. Nevertheless, in recent decades, the inclusion of FM in compound feeds has shown a clear downward trend, mainly due to the limited supplies and price fluctuations [1]. In order to sustain the growth of aquaculture industry, nutrition researchers and aquafeed producers have turned the focus to the use of alternate protein sources from terrestrial plants, which are more sustainable [2]. However, the use of plant-derived raw materials in aquafeeds has its drawbacks because of their suboptimal amino acid profile, the presence of antinutritional compounds, poor palatability, and low digestibility. In particular, limiting levels of some essential amino acids (EAAs) may lead to reduced fish growth, immunosuppression, and poor dietary efficiency.

Among EAAs, methionine (Met) is notably under-represented in vegetable meal compared to FM representing the main limiting AA in fish diets, especially in those containing high levels of soybean meal [3]. A cost-effective strategy to meet the EAAs requirement of several fish species is to fortify their diets by supplementing EAAs in a crystalline form [4,5]. However, the high leaching (dissolution) of critical components in the water and high intestinal absorption rates represent the main limits of the plant-based diets supplemented with crystalline amino acids (CAAs). Indeed, a faster absorption rate of CAAs leads to potentially higher levels of catabolism of these AAs in liver and other organs resulting in a reduced amount of AAs being available for metabolic functions, such as protein synthesis [6,7]. These metabolic consequences have been reported in recent studies, which suggest that fish utilize CAAs less efficiently than AAs supplied as intact proteins due to the asynchronous absorption [5–8].

Coating, encapsulation, microencapsulation, or polymerization are technological processes developed to increase the feed stability, overcome the loss due to leaching, and delay the intestinal absorption of CAAs [9,10].

Currently, the Met sources used to fortify aquafeeds include L-Met and its derivative forms, such as DL-Met (the racemic mixture of D- and L-isomer of Met), and Methionine Hydroxy Analogue (MHA). The main MHAs are 2-hydroxy-4-methylthio butanoic acid (HMTBa) and its calcium salt, HMTBa-Ca, both precursors of L-Met. HMTBa is a racemic mixture of Met D- and L-isomer and consists of about 65% in monomeric form, 23% in the dimer/oligomeric form, and the remaining 12% being water. Because HMTBa bears a hydroxyl group instead of an amino group, it is classified as an organic acid [10]. HMTBa and HMTBa-Ca are currently authorized for use as nutritional additives, under the functional group "amino acids, their salts and analogues".

In fish species, the bioavailability of different Met forms is different as it is mainly related to the leaching rate in the water, the intestinal uptake, and the bioconversion to L-Met. Indeed, in the study by Vázquez-Añón et al. [11], the chemical structure differences between HMTBa and DL-Met led to differences in how and where the two compounds were absorbed, enzymatically converted to L-Met and used by the animal.

The metabolic conversion of L- and D-HMTBa and D-Met to L-Met is a two-step process, with each compound being converted first to a keto-methionine intermediate called ketomethylthio-butanoic acid (KMB). The second step of conversion is a transamination of the KMB to form L-Met [11–13]. The enzymes needed for transamination are present in all tissues and the conversion to Met is very quick, such that there is no measurable pool of KMB [14]. For this reason, as a precursor of Met, HMTBa is efficacious in the promotion of growth in animals and it has been widely used as a dietary supplement in poultry, swine, and ruminants, showing positive effects on their growth [12].

In fish, knowledge on the efficacy–efficiency of different Met sources on growth performance and metabolism is scarce. In species such as rainbow trout, common carp, Nile tilapia, channel catfish, and cobia, DL-Met resulted a better source than MHA with regard to the growth performance [13,15–17]. In rainbow trout, DL-MHA calcium salt was less bioavailable (60–73%) in comparison to DL-Met based on several performance parameters, such as weight gain, growth rate, and retained nitrogen [6].

By contrast, turbot fed HMTBa and channel catfish fed MHA, MHA-Ca, and micro-capsulated DL- Met grew more than fish fed L-Met or DL-Met [18,19]. In the study by Guo et al. [20], which used a meta-analysis approach to test the effect of MHA on fish growth performance and feed utilization, a significant increase in fish production was found when MHA was properly dosed in the diet.

Differently from DL-Met and HMTBa products, L-Met is seldom used as an exogenous feed supplement, as it is obtained from DL-Met purification with high production costs. However, recently, a feed-grade L-Met product has become commercially available in some countries. The feed-grade L-Met (99% purity) can be produced from chemical synthesis, microbial fermentation of plant-based raw materials, or a combination of both [21].

DL-methionyl-DL-methionine (Met-Met) is a new dipeptide formed by the dehydration and condensation of two DL-Met molecules. Met-Met has some advantages such as better intestinal absorption than crystalline Met and lower water solubility. The latter characteristic overcomes the leaching challenges, which benefit diet efficiency, fish growth, and a healthy environment [22].

In recent years, another form of Met such as coated Met (Co-Met) has been used as exogenous feed supplement. The coating procedure has been shown to improve Met bioavailability resulting in a slower rate of breakdown and increased intestinal transition time in comparison to crystalline Met [17]. Indeed, the supplementation of microcapsulated Met rather than crystalline DL-Met in Met-deficient diets improved the growth of channel catfish (*Ictalurus punctatus*), as reported by Zhao et al. [19].

By investigating how different Met forms can influence the hepatic metabolism of carnivorous fish species, Rolland et al. [5] found that in rainbow trout, the dietary Met levels mirrored Met plasma levels and modified the expression of a wide panel of genes. Indeed, in addition to its role in protein synthesis, Met affects fatty acid oxidation, phospholipid status, creatine synthesis, bile acid production, and polyamine availability [23]. A role of Met as a signaling factor in the intermediary metabolism of rainbow trout has been reported, too [24]. In the latter study, the supplementation of Met to an insect-meal-based diet changed the transcript levels of genes involved in Met metabolism, maintaining an optimal level of the S-adenosylmethionine/S-adenosylhomocysteine (SAM/SAH) ratio in trout hepatic tissue. This is related to the role of Met as a precursor of SAM that serves as the main methyl donor in the metabolism of animals [25], including fish [26]. Furthermore, Met is the precursor of taurine (Tau) and cysteine (Cys), one of the three AAs constituting glutathione, which is the most important low-molecular-weight antioxidant synthesized in the cells [23].

Accordingly, the aim of the present work was to evaluate the effects of different exogenous Met forms supplemented to the rainbow trout diet on fish growth performance, hepatic SAM/SAH ratio levels, and mRNA copies of three genes: *SAHH* (S-adenosylhomocysteine hydrolase), *CBS* (cystathionine-beta-synthase), and *BHMT* (betaine-homocysteine methyltransferase) involved in the Met metabolism.

## 2. Materials and Methods

A fish-feeding trial was carried out at the fish facility of the Department of Agricultural, Forest, and Food Sciences (DISAFA) of the University of Torino, Italy.

The experimental protocol was designed according to the guidelines of the European Union Council 2010/63/EU for the use and care of experimental animals. The experimental protocol was approved by the Ethical Committee of the University of Torino (protocol n° 143813).

### 2.1. Experimental Diets

A positive (BAS+) and a negative (BAS−) basal-control-extruded experimental diet were prepared by VRM Naturalleva S.r.l. (Verona, Italy). The BAS+ [crude protein (CP): 49.52% as it is, ether extract (EE): 5.96% as it is] was characterized by high level of FM. BAS+ contained 1% of Met that only derived from the raw ingredients (Table 1). In contrast, BAS− diet [CP: 50.71% as it is, EE: 4.49% as it is] was characterized by high level of vegetable meals and was deficient in Met (0.7%). Additionally, the amino acid profiles of BAS+ and BAS− diets (Table 2) were reported which confirmed the dietary Met content.

At DISAFA facility, the two basal diets were ground. Ground BAS+ was mixed with fish oil (870 g and 130 g, respectively) to obtain 1 kg of finished mixture (CTRL+ diet). Water was added to the mixture and pellets were obtained using a 1 mm die meat grinder. Then, pellets were dried at 50 °C for 48 h. Similarly, ground BAS− (857 g) was mixed with fish oil (143 g) and pellets were prepared as aforementioned to obtain a CTRL− diet.

To prepare the five experimental diets, 5 g of each Met form− (Met-Met; L-Met; HMTBa; DL-Met, and Co DL-Met) was firstly mixed with fish oil and then added to BAS−

diet following the same protocol used to prepare CTRL− diet. Table 3 shows the proximate composition of the seven experimental diets.

**Table 1.** Raw materials (g/kg as it is) of basal diets.

|  | BAS+ | BAS− |
|---|---|---|
| Fish meal A [a] | 0.00 | 13.56 |
| Fish meal B [b] | 45.00 | 0 |
| Fish protein hydrolysate | 5.00 | 0 |
| Fish oil | 12.94 | 14.30 |
| Rapeseed meal | 6.70 | 0.00 |
| Soybean meal | 0.00 | 19.33 |
| Guar germ meal | 0.00 | 20.00 |
| Wheat milling | 22.52 | 8.00 |
| Corn gluten meal | 0.00 | 8.00 |
| Wheat gluten | 4.92 | 1.00 |
| Soy protein concentrate | 0.00 | 12.00 |
| Emulsifier (E484) | 0.20 | 0.20 |
| MAP [c] | 1.20 | 1.90 |
| Lysine hydrochloride | 0.30 | 1.40 |
| Vitamin and mineral premix [d] | 0.65 | 0.65 |
| Stay C 35% | 0.07 | 0.06 |
| Taurine | 0.50 | 0.60 |

Legend: [a] Fish meal A: protein content 66%; [b] Fish meal B: protein content 60%; [c] Monoammonium phosphate; [d] Vitamin and mineral premix (quantities in 1 kg of mix): Vitamin A, 4,000,000 IU; Vitamin D3, 800,000 IU; Vitamin C, 25,000 mg; Vitamin E, 15,000 mg; Inositol, 15,000 mg; Niacin, 12,000 mg; Choline chloride, 6000 mg; Calcium Pantothenate, 3000 mg; Vitamin B1, 2000 mg; Vitamin B3, 2000 mg; Vitamin B6, 1800 mg; Biotin, 100 mg; Manganese, 9000 mg; Zinc, 8000 mg; Iron, 7000 mg; Copper, 1400 mg; Cobalt, 160 mg; Iodine 120 mg; Anticaking and Antioxidant + carrier, making up to 1000 g.

**Table 2.** Amino acid composition of the basal diets: positive (BAS+) and negative (BAS−) basal diets.

| Amino Acids (% as Is) | BAS+ | BAS− |
|---|---|---|
| Ala | 2.68 | 2.08 |
| Arg | 2.50 | 2.93 |
| Asp | 3.76 | 3.72 |
| Glu | 7.45 | 6.79 |
| Gly | 2.75 | 2.30 |
| His | 0.97 | 1.05 |
| Iso | 1.97 | 1.56 |
| Leu | 3.32 | 3.11 |
| Lys | 3.06 | 3.13 |
| Met | 1.00 | 0.70 |
| Phe | 1.78 | 2.07 |
| Pro | 2.37 | 2.21 |
| Ser | 2.01 | 1.85 |
| Thr | 1.79 | 1.56 |
| Tyr | 1.38 | 1.34 |
| Trp | 0.50 | 0.39 |
| Val | 2.35 | 1.81 |
| Tau | 0.66 | 0.65 |

**Table 3.** Proximate composition (g/kg as it is) of the experimental diets.

|  | CTRL+ | CTRL− | Met-Met | L-Met | DL-Met | HMTBa | Co DL-Met |
|---|---|---|---|---|---|---|---|
| DM | 95.79 | 95.64 | 95.91 | 95.63 | 95.42 | 95.67 | 95.21 |
| CP | 44.43 | 44.19 | 44.23 | 44.83 | 44.23 | 44.73 | 44.41 |
| EE | 21.70 | 21.20 | 21.54 | 21.19 | 21.28 | 21.24 | 21.49 |
| Ash | 7.05 | 7.12 | 7.03 | 7.05 | 7.03 | 7.14 | 7.11 |

Legend: DM dry matter, CP crude protein, EE ether extract.

## 2.2. Feeding Trial and Growth Performances

A 59-day feeding trial was performed using 567 rainbow trout (*Oncorhynchus mykiss*) fingerlings. Fish were lightly anesthetized using MS-222; PHARMAQ Ltd., UK and then individually weighted (KERN PLE-N v. 2.2; KERN & Sohn GmbH, Balingen-Frommern, Germany). The mean initial body weight (iIBW) was 3.40 ± 0.40 g. After weighting, trout were randomly distributed into 21 fiberglass tanks of 100 L volume connected to a flow-through open system supplied with artesian well water at a constant temperature of 13 ± 1 °C, and flow rate of 8 L min$^{-1}$. Dissolved oxygen was measured once a week and it ranged between 7.6 and 8.7 mg L$^{-1}$.

Fish were fed manually to apparent satiety (the feed distribution was stopped as soon as the fish stopped to eat) three times a day, seven days per week. Each of the seven experimental diets was fed to 3 tanks of trout. Fish mortality was checked daily. To measure growth, fish were weighed in bulk every 2 weeks. At the end of the trial, after 24 h of fasting, fish were individually weighted and mean individual final body weight–(iFBW) was calculated.

The following growth performance indices were calculated, too:

- Mortality (%) (M%) = [(number of dead fish/number of fish at the beginning) × 100];
- Individual weight gain (iWG, g) = iFBW (individual final body weight, g) − iIBW (individual initial body weight, g);
- Feed conversion ratio (FCR) = total feed supplied [g, dry matter (DM)/WG (g);
- Protein efficiency ratio (PER) = WG (g)/total protein fed (g, DM);
- Specific growth rate (SGR, % day-1) = [(lnFBW − lnIBW)/number of feeding days] × 100;
- Feeding rate (FR, %/d) = [(total feed supplied (g, DM) × 100/number of feeding day)]/[e$^{(lnFBW+lnIBW)\times 0.5}$];
- Thermal growth coefficient = (FBW$^{1/3}$ − IBW$^{1/3}$)/Σ(T × d) × 100 (where d, is the number of days of trial).

## 2.3. Chemical Analyses of Diets and Fish Whole-Body Composition

Samples of experimental diets were finely ground with a cutting meal (MLI 204; Bühler AG, Uzwil, Switzerland) and analyzed for DM (#934.01), CP (#984.13), and ash (#942.05) contents according to AOAC International (2000). EE (#2003.05) was analyzed according to AOAC International (2003). To perform the assessment of the fish whole-body composition (WBC), at the end of the trial and after 24 h of fasting, 3 fish per tank (9 fish per diet) were sampled, euthanized by overdose of MS-222 (300 mg/L), ground with a knife mill (Grindomix GM200; Retsch GmbH, Haan, Germany), and frozen at −80 °C until analyses. For the analysis, the same methods described for feed sample analysis were used.

## 2.4. Gene Expression Analysis

### 2.4.1. Total RNA Extraction and cDNA Synthesis

Total RNA was extracted from each sample of rainbow trout liver using an automatic system (Maxwell® 16 Instrument, Promega, Milan, Italy), and a total RNA purification kit (Maxwell® 16 Tissue LEV). Briefly, 125 mg of each tissue was mixed with ice-cold Lysis Solution containing 1-thioglycerol and then homogenized using GentleMACS Dissociator (Miltenyi Biotec, Milan, Italy) until no visible tissue fragments remained. The RNA Dilution Buffer was added to the lysate; then, the whole volume was vortexed and transferred into Maxwell® 16 LEV Cartridges for RNA isolation.

The quantity of the total RNA extracted was calculated by measuring the absorbance at 260 nm using a NanoDrop 2000 c spectrophotometer (Thermo Scientific, Milan, Italy), whereas RNA integrity was verified by agarose gel electrophoresis. The purity of RNA was assessed at 260/280 nm and averaged 2.10 ± 0.01.

After extraction, 3 μg of total RNA was reverse-transcribed into cDNA in a mix of 20 μL volume containing oligo dT16 primers, as described in the SuperScript III reverse transcriptase kit (Invitrogen, Milan, Italy).

2.4.2. Generation of In Vitro-Transcribed mRNAs for Each Gene Standard Curves

Two sets of forward and reverse primers were designed based on Atlantic salmon BHMT (betaine-homocysteine methyltransferase), and SAHH (S-adenosylhomocysteine hydrolase) cDNA sequences available in the NCBI database (GenBank accession number: BT043706 and NM_001140375, respectively). Another set of primers was designed for cystathionine-beta-synthase (CBS) based on the rainbow trout cDNA sequence available in the NCBI database (GenBank acc. nr.: NM_001124686). To generate the in vitro-transcribed mRNAs for *BHMT*, *SAHH*, and CBS genes, liver cDNA was amplified via PCR, using for each gene a forward primer engineered to contain a T7 phage polymerase promoter sequence at its 5′ end and a reverse primer (Table 4). PCR products were then run on a 2% agarose gel stained with ethidium bromide. PCR products were subsequently cloned using the pGEM®-T cloning vector system (Promega, Milan, Italy) and then sequenced in both SP6 and M13 directions.

**Table 4.** Sequences of the primers used for molecular cloning and one-step quantitative real-time RT-PCR (reverse transcription polymerase chain reaction).

| Gene | Nucleotide Sequence (5′→3′) | Purpose |
|------|------------------------------|---------|
| | | Cloning |
| *BHMT FW* | TGCAGAGTACTTTGAGCACGT | |
| *BHMT RV* | CCGTGACTACTGGGAGAAGC | |
| *SAHHB FW* | CCCTTCAAAGTTGCTGACATCA | |
| *SAHHB RV* | ATGTGTGGTGCATTGAGCAGA | |
| *CBS FW* | AAACCCTGGTGGTGGAAC | |
| *CBS RV* | GTGCTCTACAAACAATTCAAACAGGT | |
| | | Standard Curve |
| *T7 BHMT sense* | gtaatacgactcactatagggTGAAAGAGGGAGTGGAGAGG | |
| *BHMT antisense* | CCGTGACTACTGGGAGAAGC | |
| *T7 SAHHB sense* | gtaatacgactcactatagggAGATGAGGGAGCTGTATGGC | |
| *SAHHB antisense* | ATGTGTGGTGCATTGAGCAGA | |
| *T7 CBS sense* | gtaatacgactcactatagggAAACCCTGGTGGTGGAAC | |
| *CBS antisense* | GTGCTCTACAAACAATTCAAACAGGT | |
| | | Real-time RT-PCR |
| *BHMT FW* | TGCCAGGGATTCATCGATCTG | Amplicon size: 75 bp; E = 91%; R2 value = 0.99 |
| *BHMT RV* | ATGACCAGGTGGGACATGCAC | |
| *BHMT probe* | AGAATTCCCCTTCGGTCTGGAGCCCA | |
| *SAHHB FW* | CCGCCGTGCTCATTGAGA | Amplicon size: 65 bp; E = 93%; R2 value = 0.99 |
| *SAHHB RV* | GTTCAATGGTCCAGCTGCAATATC | |
| *SAHHB probe* | CTGCCCTTGGAGCCGA | |
| *CBS FW* | AGACCATCAAGATCCTCAAGGAGAA | Amplicon size: 62 bp; E = 94%; R2 value = 0.99 |
| *CBS RV* | TCGTTGACGAGTCCGGC | |
| *CBS probe* | GGCTTTTGACCAGG | |

In vitro transcription was performed using T7 RNA polymerase and other reagents supplied in the RiboProbe In Vitro Transcription System kit (Promega, Milan, Italy) following the manufacturer's protocol.

The synthetic mRNAs obtained by in vitro transcription were used as quantitative standards in the real-time PCR analysis of the biological samples. For this, defined amounts of mRNAs at 10-fold dilutions were amplified via real-time PCR using iTaq™ Universal Probes One-Step kit (Bio-Rad, Milan, Italy) as reported in Terova et al. [24]. The Ct values obtained by amplification were used to create standard curves for target gene quantification.

### 2.4.3. Transcript Quantification by One-Step TaqMan® Real-Time RT-PCR

One hundred nanograms of total RNA extracted from each trout liver was subjected to One-step Taqman® quantitative real-time RT-PCR. Biological samples were loaded in the same 48-well plate with standard mRNAs and run under the same aforementioned real-time RT-PCR conditions.

Real-time Assays-by-Design^SM PCR primers and gene-specific fluorogenic probes were designed by Invitrogen, Milan, Italy.

Primer sequences and TaqMan® probes used for each target gene amplification are shown in Table 3. TaqMan® PCR reactions were performed on a Bio-Rad® CFX96. System.

### 2.5. SAM/SAH HPLC Analysis

#### 2.5.1. Standards and Sample Processing

For the HPLC analysis, SAM and SAH standards (Sigma, St. Louis, MO, USA) were first prepared in HPLC water at 1 mM concentration and then sequentially diluted with 0.4 M perchloric acid ($HClO_4$). To ensure a good calibration curve, five concentrations were prepared.

To extract SAM and SAH from biological samples, 0.2 g of fish liver (wet tissue) was mixed with 800 µL of perchloric acid (0.4 M) and homogenized for 4 min using a TissueLyser II (Qiagen, Milan, Italy), at 4 °C with stainless steel beads of 5 mm diameter. After centrifugation at $10,000\times g$, for 15 min at 4 °C, the supernatant was collected and then filtered into vials through a 0.22 µm syringe filter. The extracted solution was stored at −80 °C until HPLC analysis.

#### 2.5.2. HPLC System and Chromatographic Conditions

The HPLC system consisted of a PU-2089 quaternary pump connected to a degasser and a MD-2015 diode array, both obtained from Jasco-Europe S.r.l Company (Milan, Italy). Separation was carried out in a Kinetex XB-C18 reversed-phase analytical column (250 mm × 4.6 mm, length × internal diameter) with a particle size of 5 µm (Phenomenex, Milan, Italy) equipped with an HPLC universal guard column (Phenomenex, Milan, Italy). Solvent A and B formed the gradient elution.

Solvent A was an aqueous buffer containing 8 mM octanesulfonic acid sodium salt and 50 mM $NaH_2PO_4$, and it was adjusted to pH 3 with $H_3PO_4$. Solvent B consisted of 100% methanol. Solvent A was filtered through a 0.2 µm membrane filter. The chromatographic conditions are reported inTerova et al. [24]. Data processing was performed using ChromNAV (Jasco-Europe s.r.l.). SAM and SAH quantifications were performed by comparing each peak area with the peak area of the calibration curve.

#### 2.5.3. Validation

For the validation, each biological sample and each standard were run three times. A blank sample was injected daily at the beginning of analysis to check the baseline and to monitor the matrix effect due to sample injection. A blank was also run between each sample to ensure there was no memory.

The equation of the calibration curves showed $r^2 = 0.998$ for SAM, and $r^2 = 0.998$ for SAH.

### 2.6. Statistical Analyses

The growth performance and fish whole-body composition were analyzed by one-way ANOVA (IBM SPSS Statistics v. 25.0 for Windows) using the following model:

$$Yij = \mu + Di + \varepsilon ij$$

where Yij = observation; µ = overall mean; and Di = effect of diet; εij = residual error.

The assumption of normality was checked using the Shapiro–Wilk test. Levene's homogeneity of variance test was used to assess homoscedasticity. If such an assumption

did not hold, the Brown–Forsythe statistic was applied to test the equality of group means instead of the F one. Pairwise multiple comparisons were performed to test the difference between each pair of means (Tukey's test and Tamhane's T2 in the case of assumed or not assumed equal variances, respectively).

Met metabolism and the gene expression data were analyzed by one-way ANOVA using GraphPad Prism 8 software (San Diego, CA, USA). If there were significant differences between dietary groups, Tukey's test was applied as post hoc analysis.

The results were expressed as the mean and pooled standard error of the mean (SEM). Significance was set at $p \leq 0.05$.

## 3. Results

### 3.1. Fish Performance

The mortality and growth performance parameters are reported in Table 5. Fish readily accepted the experimental diets and at the end of the trial, no statistical differences were found for any of the performance parameters considered.

**Table 5.** Mortality (%) and growth performance parameters of rainbow trout fed experimental diets (n = 3; mean ± SD).

|  | CTRL+ | CTRL− | Met-Met | L-Met | DL-Met | HMTBa | Co DL-Met | SEM |
|---|---|---|---|---|---|---|---|---|
| M (%) | 3.70 ± 3.70 | 9.88 ± 2.14 | 6.17 ± 2.14 | 7.41 ± 5.24 | 8.64 ± 4.28 | 7.41 ± 3.07 | 6.17 ± 5.66 | 2.09 |
| iIBW (g) | 3.40 ± 0.03 | 3.41 ± 0.03 | 3.39 ± 0.02 | 3.40 ± 0.05 | 3.40 ± 0.02 | 3.41 ± 0.04 | 3.40 ± 0.03 | 0.01 |
| iFBW (g) | 21.40 ± 1.73 | 20.18 ± 1.38 | 20.05 ± 1.42 | 19.31 ± 1.72 | 20.66 ± 1.29 | 20.57 ± 2.23 | 20.99 ± 1.75 | 0.34 |
| WG (g) | 18.00 ± 1.76 | 16.77 ± 1.35 | 16.67 ± 1.42 | 15.91 ± 1.67 | 17.26 ± 1.23 | 17.17 ± 2.19 | 17.59 ± 1.72 | 0.33 |
| FCR | 0.82 ± 0.07 | 0.87 ± 0.11 | 0.82 ± 0.01 | 0.82 ± 0.03 | 0.81 ± 0.03 | 0.82 ± 0.090 | 0.78 ± 0.01 | 0.01 |
| PER | 2.87 ± 0.25 | 2.75 ± 0.34 | 2.83 ± 0.03 | 2.83 ± 0.11 | 2.93 ± 0.12 | 2.86 ± 0.30 | 3.02 ± 0.02 | 0.04 |
| SGR (%/d) | 3.13 ± 0.312 | 2.61 ± 0.59 | 2.97 ± 0.12 | 2.84 ± 0.06 | 2.98 ± 0.15 | 2.88 ± 0.20 | 3.04 ± 0.06 | 0.06 |
| FR (%/d) | 2.94 ± 0.16 | 2.47 ± 0.39 | 2.76 ± 0.12 | 2.62 ± 0.03 | 2.74 ± 0.06 | 2.65 ± 0.13 | 2.71 ± 0.07 | 0.04 |
| TGC | 0.16 ± 0.01 | 0.15 ± 0.01 | 0.15 ± 0.01 | 0.15 ± 0.011 | 0.16 ± 0.01 | 0.15 ± 0.01 | 0.16 ± 0.01 | 0.01 |

Table 6 reports the fish WBC at the end of the trial. All parameters were affected by the dietary treatment. Fish fed the Co DL-Met diet showed higher DM content than fish fed the Met-Met diet. CP was the lowest in CTRL+. Fish fed CTRL− had a higher EE compared to fish fed L-Met and Met-Met, whereas fish fed Met-Met had the highest ash content.

**Table 6.** Whole-body proximate (g/100 g, as it is) composition of rainbow trout fed the experimental diets (n = 9). PG and EG values were obtained as estimates. Different superscript letters indicate statistical significance.

|  | CTRL+ | CTRL− | Met-Met | L-Met | DL-Met | HMTBa | Co DL-Met | SEM | p |
|---|---|---|---|---|---|---|---|---|---|
| DM | 25.75 ± 0.52 [ab] | 26.62 ± 0.79 [ab] | 25.66 ± 0.40 [b] | 26.61 ± 0.56 [ab] | 26.28 ± 0.41 [ab] | 26.59 ± 0.19 [ab] | 26.71 ± 0.88 [a] | 0.10 | 0.009 |
| CP | 13.97 ± 0.31 [c] | 14.74 ± 0.61 [abc] | 14.18 ± 0.54 [bc] | 15.07 ± 0.70 [ab] | 14.79 ± 0.54 [abc] | 15.35 ± 0.35 [a] | 15.28 ± 0.57 [a] | 0.11 | 0.000 |
| EE | 8.47 ± 0.43 [abc] | 9.15 ± 060 [a] | 7.93 ± 0.27 [c] | 8.36 ± 0.36 [bc] | 8.54 ± 0.32 [abc] | 8.83 ± 0.71 [ab] | 8.66 ± 0.44 [abc] | 0.08 | 0.001 |
| Ash | 2.99 ± 0.12 [b] | 2.26 ± 0.09 [c] | 3.25 ± 0.061 [a] | 2.21 ± 0.11 [c] | 2.22 ± 0.16 [c] | 2.16 ± 0.06 [c] | 2.35 ± 0.12 [c] |  | 0.000 |
| PG | 2.49 ± 0.07 [bcd] | 2.48 ± 0.10 [bcd] | 2.35 ± 0.11 [d] | 2.41 ± 0.13 [cd] | 2.56 ± 0.11 [abc] | 2.66 ± 0.07 [ab] | 2.71 ± 0.13 [a] | 0.07 | <0.001 |
| EG | 1.74 ± 0.09 [a] | 1.77 ± 0.13 [a] | 1.51 ± 0.05 [a] | 1.54 ± 0.07 [bc] | 1.69 ± 0.07 [ab] | 1.74 ± 0.09 [a] | 1.74 ± 0.06 [a] |  | <0.001 |

Abbreviations: DM, dry matter; CP, crude protein; EE, ether extract; PG, protein gain; EG, ether gain.

### 3.2. Hepatic SAM and SAH Levels

As shown in Figure 1A, SAM was significantly higher in the L-Met dietary group than in the following dietary groups: CTRL− ($p < 0.05$); Met-Met ($p < 0.05$); HMTBa ($p < 0.05$); DL-Met ($p < 0.05$); and Co DL-Met ($p < 0.05$). A similar trend was reported for SAH (Figure 1B), whose level in the L-Met dietary group was significantly higher with respect to the CTRL− group ($p < 0.05$). Moreover, the L-Met dietary group displayed a significantly higher level of SAH with respect to the HMTBa ($p < 0.05$), Met-Met ($p < 0.05$), and Co DL-Met ($p < 0.05$) dietary groups. The CTRL− dietary group had significantly lower levels of SAH with respect to CTRL+ ($p < 0.05$) and DL-Met ($p < 0.05$). The significance level of

0.05 was also detected for the comparisons of the CTRL− vs. Met-Met dietary group and CTRL− vs. Co DL-Met dietary group, respectively.

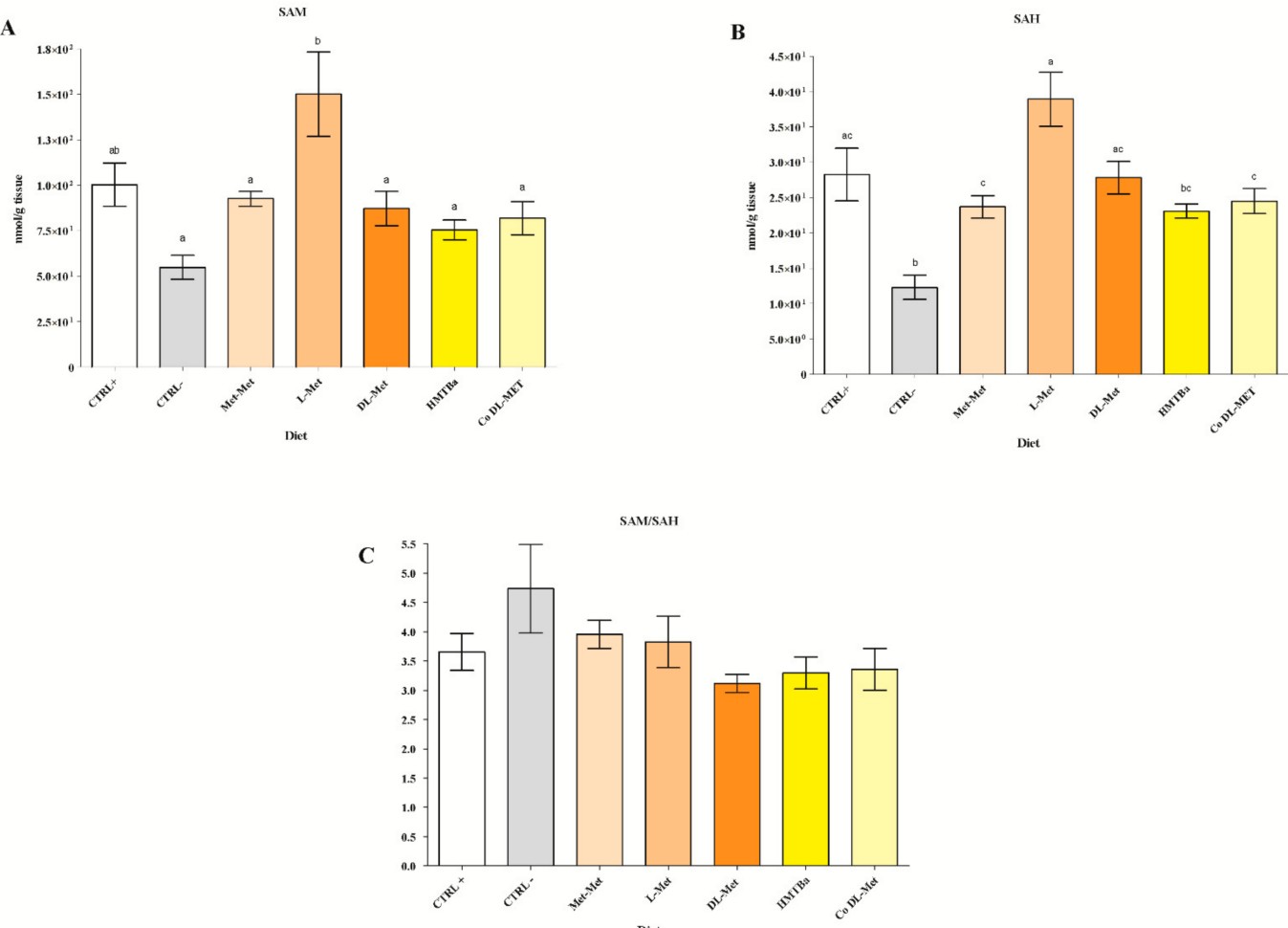

**Figure 1.** SAM and SAH concentrations (nmol/g) (**A**,**B**) and the SAM/SAH ratio (**C**) in liver of rainbow trout. Data are presented as mean ± SEM (n = 6). Different letters denote statistical significance ($p < 0.05$).

The SAM/SAH ratio showed no statistical differences among the diets (Figure 1C).

*3.3. Expression of Genes Involved into Met Metabolism*

As shown in Figure 2A, an upregulation of the *BHMT* transcript level ($p < 0.05$) was found in fish fed with L-Met in comparison to fish fed with a Met deficient diet (CTRL−) and with DL-Met, and HMTBa-supplemented diets.

The mRNA copies of the *CBS* gene (Figure 2B) were significantly lower in the CTRL+ dietary group ($p < 0.05$) than in the HMTBa and Co DL-Met groups.

The *SAHH* mRNA copy number did not show differences related to diets being at similar levels in all the dietary groups (Figure 2C; $p > 0.05$).

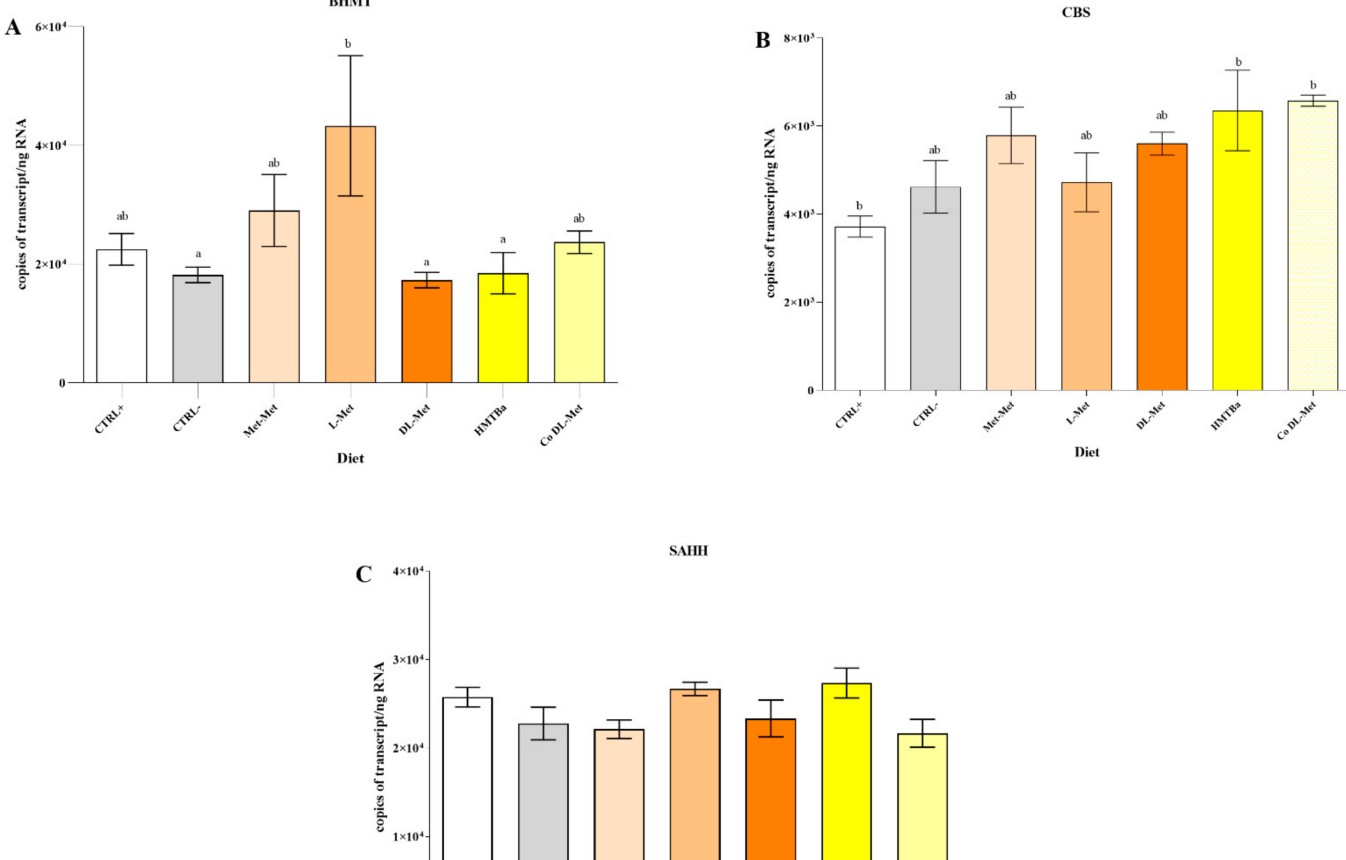

**Figure 2.** Transcript copies of genes coding for *BHMT*, *CBS*, and *SAHH* enzymes (**A**–**C**) as quantified in the liver tissue of rainbow trout fed either a control diet deficient in Met (CTRL−), a positive control diet (CTRL+), or five diets supplemented with different types of Met (Met-Met; L-Met; HMTBa; DL-Met, and Co DL-Met). Data are represented as mean ± SEM (n = 6). Different letters denote statistical significance ($p < 0.05$).

## 4. Discussion

The use of synthetic Met sources is a prerequisite to correct the Met deficiency in plant-based diets and allow the normal and healthy growth of fish. There is limited information and debates still exist on the bioavailability of different Met sources, which mainly depends on their intestinal absorption, gut microbial metabolism, and the conversion of isomers into the biologically active form of L-Met [15]. For instance, Met derivatives, such as DL-Met and MHA, have shown variable bioavailability in comparison to L-Met. In rainbow trout, MHA was absorbed at a slower rate than DL-Met, because it was more ready to be used by gut microbiota than DL-Met, either as a preferential substrate or being less transported by the intestine, leaving it to be used by bacteria [13,15].

Accordingly, our research focused on the bioefficacy of five dietary Met sources in a commercially relevant species, rainbow trout. Five experimental diets supplemented with different forms of Met (Met-Met; L-Met; HMTBa; DL-Met; and Co DL-Met) were tested in a two-month feeding trial. In addition, a CTRL+ diet with a high level of FM and 1% of Met deriving only from the raw ingredients, and a CTRL− diet characterized by a high level of vegetable meal and deficiency in Met, were tested too.

With regard to fish performance indexes, at the end of the feeding trial, no differences were found between the Met-supplemented dietary groups and negative and positive

control groups. Mortality ranged from 3.70% (CTRL+) to 9.88% (CTRL−) and, even if it was numerically higher in the CRTL− group, no statistical differences between groups were recorded. At the end of the trial, fish of all dietary groups reached an FBW about 6-fold higher than the IBW, suggesting an optimal nutrient utilization. This is also sustained by the low FCR, which was found to be less than 0.9 in all dietary groups, as well as by the high PER and SGR values.

In rainbow trout, several trials have reported differences in bioefficacy between MHA and DL-Met based on the growth parameters. Powell et al. [7] found a lower bioefficacy of MHA with respect to DL-Met in juvenile rainbow trout, whereas in our study, fish fed MHA- and DL-Met-supplemented diets grew similarly well. In agreement with our results, the performance indexes of red drum fed with MHA-, Alimet® (MHA liquid form)-, and DL-Met-supplemented diets were statistically not different from fish fed with L-Met-supplemented diets [27]. The rainbow trout in our study was capable of using the various Met compounds as effectively as L-Met to meet their Met demand. Since the Met requirement was met, an additional supplementation of Tau in the diet could have revealed a potential positive effect on growth parameters [28]. However, in the study by Boonyoung et al. [28], dietary Tau addition (0.6 g/kg) in combination with MHA as a Met source, did not improve growth performance when dietary Met levels met the requirements of rainbow trout fed with a soybean protein concentrate diet.

As known, Tau is an amino sulfonic acid widely distributed in animal tissues, but very low in plants [29]. For this reason, it is used as feed additive in plant-based diets. Tau has a role as a growth promoter [30] and it is involved in various key biological functions in fish, including conjugation of bile salt, osmoregulation, neurotransmitter, stabilization of cell membranes, and scavenging of reactive oxygen species [31]. The freshwater species, such as rainbow trout, can synthesize Tau from cysteine (Cys) and Met due to the high hepatic activity of cysteine sulfinate decarboxylase. Therefore, Tau was not necessary for rainbow trout when fish were fed with a Met-supplemented diet [28,29]. Anyway, the metabolism of sulfur-containing amino acids and the mechanism of action of different Met sources are yet to be further elucidated.

Body composition, specifically crude protein, moisture, ash, and lipid content of carcasses, was significantly affected by Met sources. Fish fed the CTRL+ diet had a lower crude protein content than fish fed the Met analogue (HMTBa) and Co DL-Met. It seems that these two Met forms promoted protein accretion more efficiently than a diet in which Met was supplied as intact protein. In common carp [16] and in rainbow trout [6], DL-Met showed higher bioavailability and promoted more the protein synthesis than HMTBa. This is partially in contrast with our data, which showed that trout can effectively utilize the HMTBa, too, suggesting that trout can effectively convert the D-isomer of Met into the L-isomer.

In our feeding trial, the coated form of DL-Met performed better than DL-Met. The coating technique most likely delayed the leaching of AA and allowed Met to be slowly released in the intestine. In this way, free Met supplemented to the feed was absorbed synchronously with Met derived from dietary protein digestion [3]. The coating material consists of natural materials, such as stearic acid, glyceride, and gelatin, that are difficult to be degraded by gut microbiota and sensitive to the change in pH; these characteristics help DL-Met to be continuously released in the intestine and effectively be used in protein synthesis [3,9]. The two crystalline forms, L-Met and DL-Met, performed similarly in our feeding trial as they share the same mechanism of absorption in fish. Indeed, CAAs are absorbed faster and assimilated more efficiently than protein, which is broken down into smaller chains of amino acids before absorption [32].

With regard to the lipids, the highest body lipid content in our study was found in trout that received the Met-deficient diet, whereas both protein-bound and synthetic Met (Met-Met, L-Met) promoted a significant body lipid reduction as a result of an efficient utilization of these forms by fish.

Nearly half of the Met metabolism (48%) and up to 85% of all methylation reactions take place in the liver [33]. Therefore, dietary and hepatic Met levels are tightly associated.

The activated Met, namely SAM, is an important methyl donor and it is involved in several reactions, such as the methylation of DNA, RNA, and proteins. Following the transfer of a methyl group to an acceptor molecule, SAM is converted to SAH, which is a potent competitive inhibitor. Consequently, elevated levels of SAH suppress SAM-dependent transmethylations and have an effect upon the ratio of SAM/SAH within hepatic tissue. The SAM/SAH ratio is influenced by the activity of S-adenosylhomocysteine hydrolase (SAHH), which catalyzes the reversible hydrolysis of SAH to Hcy and adenosine.

Hcy is an important intermediate of Met metabolism and by the action of BHMT enzyme, the methyl group of betaine molecule can be transferred to Hcy and regenerate Met. The Met pool can also be used in the trans-sulfuration pathway producing Cys and Tau in reactions requiring cystathionine β-synthase (CBS) [33].

In our study, the *CBS* gene was downregulated in fish fed the CTRL+ diet, in which Met was supplied by raw ingredients. The CTRL− diet did not promote any significant changes in *CBS* gene expression, but when CTRL− to CTRL+ dietary groups are compared, although not statistically significant, the low-Met diet numerically promoted an increase in CBS mRNA copies. This trend was also found in salmonids fed a Met-deficient diet; in those fish, the *CBS* gene expression increased [34,35].

The two Met sources, HMTBa and Co DL-Met, increased the mRNA copies of *CBS* gene in the liver tissue. In agreement with our results, Zuo et al. [36] found that HMTBa, but no other Met sources such as L-Met, significantly increased the transcript level of *CBS* in hepatic cells from porcine (IPEC-J2 cells), thus favoring the trans-sulfuration. In chicken small intestine, HMTBa (DL-HMTBa) was preferentially diverted to the trans-sulfuration pathway, thus leading to an increase in Cys and Tau content in the enterocytes [37]. As already mentioned, Met is also precursor of Tau and Cys, one of the three AAs constituting glutathione (γ-l-glutamyl-l-cysteinyl-glycine), which is the most important low-molecular-weight antioxidant synthesized in the cells. Therefore, HMTBa may maintain high levels of antioxidants in the cells by promoting Met trans-sulfuration.

Our data show that the Met-deficient diet reduced significantly *BHMT* mRNA copies in trout liver. The same downregulation was found in both whole fry and broodstock liver of rainbow trout by Fontagné-Dicharry et al. [34]. On the contrary, the remethylation pathway was upregulated in fish fed with L-Met to meet the cellular demand for Met. The Met free form, namely L-Met, contributed to an increase in the intracellular Met concentration, and it could have been directly used to synthetize SAM.

In this regard, feeding rats with increasing levels of Met increased hepatic SAM levels, as a result of higher hepatic methionine adenosyl transferase (MAT) activity [38]. This enzyme responds to the Met supply by catalyzing the biosynthesis of SAM from L-Met and ATP. We did not evaluate the expression of the *MAT* gene encoding for the corresponding enzyme. Nevertheless, we can postulate that the liver of fish fed the L-Met-supplemented diet had a higher methyl transfer capacity due to the increased SAM levels. Furthermore, SAH hepatic concentration significantly increased in trout fed L-Met with respect to fish fed other Met sources. A similar metabolic response was found in Atlantic salmon liver, in which Met intake affected the methylation capacity [39]. As a matter of fact, Espe et al. [39] did not find significant differences in either feed intake or growth. Neither carcass protein nor lipid retention was affected by methionine intake. However, Espe et al. [39] concluded that high methionine intake was essential to secure high synthesis of activated methyl groups, maintaining liver health while increasing the hepatic taurine production keeping the hepatic-free methionine constant at all intakes. Our findings confirmed the high bioefficacy of L-Met being easily absorbed at the intestinal level and affecting hepatic Met metabolism, in particular, the methylation–remethylation pathways.

The Co DL-Met and HMTBa affected the trans-sulfur pathway more than the other Met sources, favoring the production of antioxidants. Indeed, the Co-DL Met and HMTBa diets increased the transcript levels of the *CBS* gene over the levels found in fish fed with

the CTRL+ diet, maintaining a positive health status of rainbow trout. This was also highlighted by the high CP content recorded by the Co-DL Met and HMTBa feeding groups with respect to CTRL+.

In general, all tested Met forms did not influence the methylation ratio (SAM/SAH) as confirmed by the similar expression of the *SAHH* gene in fish fed with different diets. We can hypothesize an Hcy accumulation at the hepatic level without compromising fish health. Future studies should quantify the levels of Met and Hcy in liver.

In summary, the present study showed an optimal dietary intake of all tested Met sources with similar promoting effects on fish growth, and hepatic Met metabolism. Nevertheless, the mechanisms underlying these effects warrant further investigation.

**Author Contributions:** Conceptualization, G.T. and L.G.; methodology, C.C. (Chiara Ceccotti); I.B., L.G., C.C. (Christian Caimi), S.B.O., S.R. and F.B.; Data collection, curation, and analysis, C.C. (Chiara Ceccotti), I.B., L.G., C.C. (Christian Caimi), S.B.O. and S.R.; writing—original draft preparation, C.C. and G.T.; writing—review and editing, C.C. (Chiara Ceccotti), L.G., C.C. (Christian Caimi) and G.T.; funding acquisition, G.T. All authors have read and agreed to the published version of the manuscript.

**Funding:** This research was partially funded by AGER, Network Foundation, Project Fine Feed for Fish (4F), Rif. No. 2016-01-01. This work was also cofunded by the EU Horizon 2020 AquaIMPACT (Genomic and nutritional innovations for genetically superior farmed fish to improve efficiency in European aquaculture), number: 818367.

**Institutional Review Board Statement:** The animal study protocol was approved by the Bioethics Committee of University of Torino (protocol nr. 143813 and date of approval 19 March 2019).

**Informed Consent Statement:** Not applicable.

**Data Availability Statement:** Not applicable.

**Acknowledgments:** The authors would like to thank the VRM S.r.l. Naturalleva company for having provided the experimental feeds.

**Conflicts of Interest:** The authors declare no conflict of interest. The funders had no role in the design of the study; in the collection, analyses, or interpretation of data; in the writing of the manuscript, or in the decision to publish the results.

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
