# Peer review of "How Different Dietary Methionine Sources Could Modulate the Hepatic Metabolism in Rainbow Trout?"

_cimb, doi:10.3390/cimb44070223_

Round 1

Reviewer 1 Report

This manuscript presents relevant information about the different forms of methionine available in the market to supplement current diets for most cultivated species. As discussed by the authors, the substitution of marine ingredients such as fishmeal with vegetable sources to promote the industry's sustainability and avoid price fluctuations and the limited supply is a common practice in aquaculture. The introduction and design of the trial are correct, but there are inconsistencies that the authors need to address. The focus of this study is to understand if methionine type supplementation impacts hepatic metabolism. Looking at the design and the results, the amino acid profile of the diets is crucial, and you should at least find some phenotypic differences among test groups. It is arguable if the authors need to present the amino acid profile since some authors do not give it, but it is essential in light of the results presented here. This is evident when the authors discuss the effect of the different methionine forms against the negative control that was supposedly deficient in methionine. There is no difference between this treatment in growth performance or gene expression of liver S-adenoylhomocysteine hydrolase, cystathionine-beta-synthase, and betaine-homocysteine methyltransferase. I will highlight these issues further down the review.

Introduction:  

Line 30: delete "the" in "the cultured"

Line 56: Please note that you have some references with author-year and others with numbering. Please go through the text and correct it for the journal format.

Line 59: The same as in line 56.

Line 71: The same as in line 56.

Line 77: The same as in line 56.

Line 90: The authors already established that are different types of  MHA. Please provide the source in both studies.

Line 111: The same as in line 56.

Line 113: The same as in line 56.

Line 168: Substitute "Mean initial body weight (iIBW) resulted 3.40 ± 0.40 g" by "The mean initial body weight (iIBW) was 3.40 ± 0.40 g".

Line 245: As a suggestion, next time, I would recommend measuring the protein content of the extract to be sure that the differences recorded are not a result of the different protein content extracted. This is standard procedure in enzymatic assays, and since you do not have the protein content of the liver would be advisable.

Line 358: With this type of set-up and results, you can not be sure if the compounds are more or less available because the authors did not record methionine digestibility. It depends on how you define the availability of a nutrient/additive. In To et al. 2020, the team collected different parts of the intestine and tested the influx of the different types of methionine in vitro, suggesting intestinal transport of DL-MHA was lower than DL-Met. Even with their design, the authors only suggested it because they were aware of the differences between in vitro and in vivo models.

Line 373: The same as in line 56. Powel et al. (7) call it bioavailability, but I agree with the authors of the present manuscript; it is bioefficacy because they measure the outcome (protein retention, weight gain, etc).  

Line 382: The same as in line 56.

Line 386 -line395: I would recommend deleting the entire paragraph as the authors do not find any link with the present manuscript or discuss it in light of the results obtained, making this information irrelevant for the discussion.

Line 390: The authors are entitled to cite their work, but in such a broad subject, you should mention a review or cite the specific authors that study the effect as a growth promoter, osmoregulation etc.  

Line 399: Protein accretion and whole-body protein content are related, but if one is higher does not mean that you have higher protein accretion. The protein content in fish is very stable, with fat and water content varying inversely. Despite the differences observed in protein content, PER was similar among the groups. You always have to consider how relevant are biologically these differences found in protein analysis. If you want to evaluate protein accretion, you should calculate protein gain. The authors should consider why the CTR- did not record lower protein accretion or growth since methionine was deficient, and you did not observe that.

Interestingly, fish fed the met-met and L-met recorded the lowest protein accretion. Nonetheless, if you run stats on protein accretion, you should not find any differences, but this is just speculation from my side since you did not present these data. The authors should give this result and the amino acid composition of the diets.

Parameters

CTRL+

CTRL-

Met-

L-Met

DL-Met

HMTBa

Co

Met

DL-Met

iIBW (g)

3.4

3.41

3.39

3.4

3.4

3.41

3.4

iFBW (g)

21.4

20.18

20.05

19.31

20.66

20.57

20.99

CP final

13.97

14.74

14.18

15.07

14.79

15.35

15.28

Cpinitial

14

14

14

14

14

14

14

Protein gain

2.51

2.50

2.37

2.43

2.58

2.68

2.73

I calculated protein gain assuming the same initial protein content.

Line 405-415: The authors should add "most likely" since they did not record leaching and rephrase the sentence because growth performance was similar among treatments. In the following sentence, the authors compare the free DL-methionine with the coated form but neglect that there was also L-met in the trial, and no differences were recorded between these two crystalline amino acids. Please elaborate on possible differences in absorption and utilisation of these two forms.  

Line 416-419: The same as for protein. I give the same example :

Parameters

CTRL+

CTRL-

Met-

L-Met

DL-Met

HMTBa

Co

Met

DL-Met

iIBW (g)

3.4

3.41

3.39

3.4

3.4

3.41

3.4

iFBW (g)

21.4

20.18

20.05

19.31

20.66

20.57

20.99

Cfat final

8.47

9.15

7.93

8.36

8.54

8.83

8.66

C fat initial

7

7

7

7

7

7

7

Fat gain

1.57

1.61

1.35

1.38

1.53

1.58

1.58

I would advise the authors to present the results as such. Most likely, you will not find any significant differences as the values are very similar. It would be a considerable improvement to the manuscript to include the composition of the liver, if possible since you discuss lipid metabolism afterwards.

Line 451: The same as in line 56.

Line 434- 439: The authors should consider analysing the diets' amino acid content. The Ctr+ group exhibits the CBS gene downregulated and the CTR- similar to the other groups. Moreover, Zao et al.also observed that while the CBS gene was up-regulated, SAM levels were lower in that group, which makes sense.

Line 463: Delete "too".

Line 464-465: The results suggest a higher bioefficacy than bioavailability. However, growth performance was similar between treatments, even with the negative control. Moreover, the authors do not provide HSI or liver composition, which would have improved the manuscript greatly from my perspective. Espe et al. 2008 did not find significant differences in either feed intake or growth. Neither carcass protein or lipid retention was affected by methionine intake. However, Espe et al 2008 concluded that high methionine intake was essential to secure high synthesis of activated methyl groups, maintaining liver health while increasing the hepatic taurine production keeping the hepatic free methionine constant at all intakes.

In conclusion, the authors should reformulate the results and, if possible, present the amino acid profile of the diets. The discussion must be improved.  

Reviewer 2 Report

This is a comparative and informative article modulation of rainbow trout by different dietary methionine sources. It is recommended to add the results in the supplementary statistics section and to compare the CTRL+ group with the CTRL- group. The other parts of the content where problems were found are described below

#1. The number of the tables were wrong.

#2. The formatting for the footnotes of the table should be adjusted.

#3. In the P6 Line 241, the formatting of bottom half of the table 5 is difficult to read

#4. In the P7 Line 255, the heading of Standards and sample processing is wrong

#5. In the P7 Line 261, 800 μL of liquid is very diffcult to centrifugate, please check this part of statement. It is recommended to increase the sample size of one centrifugation in experiment to avoid the interference of experimental errors.

#6. In the P8 Line 309, it is recommended to add standard deviations to the statistics so that the variation of each group of experiments can be observed more clearly. For instance, the mortality of CTRL- (9.44) is higher than other groups, but there is no significant difference overall. This may be due to the large standard deviation within the group. This question also appears in P12 Line 366-368.

#7. In the P9 Line 317, it is recommended to add standard deviations to the statistics in Table 2. The comparison of Table 1 shows that the differences between each group are similar to those in Table 1, but there are significant differences between the groups in this table.

#8. In the P10 Line 331, it is a bit difficult to read the representation of significant differences between samples in Figure 1. Should we consider using the different letters like a,b,c, to indicate the difference between samples?

#9. In the P10 Line 335, it showed "fish fed with L-Met in comparison to fish fed with a Met deficient diet (CTRL-) and with DL-Met, and HMTBa supplemented diets". But with the error bar on the picture it looks like that there is significant different between the L-Met and the  (CTRL+). Please confirm the statistical results in this section.

#10. In the P12 Line 396-397, it showed that "Fish fed the CTRL+ diet had a lower crude protein content than fish fed Met analogue (HMTBa) and Co DL-Met. It seems that these two Met forms have promoted protein accretion more efficiently than a diet in which Met was supplied as intact protein". However, there is no significant difference in the CP between the HMTBa groups of the CTRL-diet which lacking Met. There is a contradiction between this and the previous inference, please further explain the data in this section.

#11. In the P14 Line 476-477, it showed "all tested Met 476 sources with similar promoting effects on fish growth". This is evidenced by the fact that the weight gain values for the CTRL-group in Table 1 are lower than the values for the other groups. However, the performance of the CBS gene, which is an indicator, is higher in the CTRL- group than in the CTRL+ group. This phenomenon is also seen in Table 2, where the CP value of CTRL- is also higher than that of CTRL+ group. Does this help to explain that the CTRL- group without MET also has some higher indexes? Please further explore and explain this results of the manuscript.

#12. It is recommended to add the newer published references in the part of the conclusions.
